# Seasonal Influenza Vaccine Effectiveness in Persons Aged 15–64 Years: A Systematic Review and Meta-Analysis

**DOI:** 10.3390/vaccines11081322

**Published:** 2023-08-04

**Authors:** João Paulo Martins, Marlene Santos, André Martins, Miguel Felgueiras, Rui Santos

**Affiliations:** 1Escola Superior de Saúde, Instituto Politécnico do Porto, Rua Dr. António Bernardino de Almeida, 4200-072 Porto, Portugal; mes@ess.ipp.pt; 2CEAUL—Centro de Estatística e Aplicações, Faculdade de Ciências, Campo Grande, Universidade de Lisboa, 1749-016 Lisboa, Portugal; mfelg@ipleiria.pt (M.F.); rui.santos@ipleiria.pt (R.S.); 3Centro de Investigação em Saúde e Ambiente, Instituto Politécnico do Porto, Rua Dr. António Bernardino de Almeida, 4200-072 Porto, Portugal; andre20.martins@gmail.com; 4Escola Superior de Tecnologia e Gestão, Instituto Politécnico de Leiria, Campus 2, Morro do Lena—Alto do Vieiro, Apartado 4163, 2411-901 Leiria, Portugal

**Keywords:** influenza, test-negative design, clinical trials, efficacy, effectiveness, strains

## Abstract

Influenza is a respiratory disease caused by the influenza virus, which is highly transmissible in humans. This paper presents a systematic review and meta-analysis of randomized controlled trials (RCTs) and test-negative designs (TNDs) to assess the vaccine effectiveness (VE) of seasonal influenza vaccines (SIVs) in humans aged 15 to 64 years. An electronic search to identify all relevant studies was performed. The outcome measure of interest was VE on laboratory-confirmed influenza (any strain). Quality assessment was performed using the Cochrane risk-of-bias tool for RCTs and the ROBINS-I tool for TNDs. The search identified a total of 2993 records, but only 123 studies from 73 papers were included in the meta-analysis. Of these studies, 9 were RCTs and 116 were TNDs. The pooled VE was 48% (95% CI: 42–54) for RCTs, 55.4% (95% CI: 43.2–64.9) when there was a match between the vaccine and most prevalent circulating strains and 39.3% (95% CI: 23.5–51.9) otherwise. The TNDs’ adjusted VE was equal to 39.9% (95% CI: 31–48), 45.1 (95% CI: 38.7–50.8) when there was a match and 35.1 (95% CI: 29.0–40.7) otherwise. The match between strains included in the vaccine and strains in circulation is the most important factor in the VE. It increases by more than 25% when there is a match with the most prevalent circulating strains. The laboratorial method for confirmation of influenza is a possible source of bias when estimating VE.

## 1. Introduction

Influenza is a respiratory disease resulting from infection with the influenza virus. It is more prevalent during cold periods, with the peak of infections between November and April in the Northern Hemisphere and between June and October in the Southern Hemisphere. The influenza virus is highly transmissible in humans [1]. The World Health Organization (WHO) estimates that there are 1 billion cases of influenza worldwide each year, of which 3–5 million are severe cases [2]. An estimated 650,000 deaths per year result from influenza infection [3]. The most effective way to prevent influenza infection is through vaccination [4]. Seasonal flu vaccination campaigns represent a major investment for countries and governments. It is therefore important to assess the effectiveness of the vaccine.

The two main types of studies used to assess the seasonal influenza vaccine (SIV) performance are randomized controlled trials (RCTs) and observational studies. Among these, the most used are cohort studies and, mainly, the case-control study, known as test-negative design (TND) [5]. RCTs are always conducted for the marketing authorization of the vaccine regardless of the year in question [6]. Vaccine performance is determined by vaccine efficacy (VER), which is equal to
(1)$$\mathrm{VER}=\left(1-\mathrm{RR}\right)\times 100$$
where RR is the relative risk. These trials are very expensive and time-consuming [7]. Therefore, the application of (1) is not a parsimonious method for real-time monitoring of the efficacy of a SIV.

For monitoring the annual effectiveness of different vaccines, TND is used when laboratory confirmation is required [8]. The TND sample is composed of individuals with influenza-like illness (ILI) who access a hospital or other healthcare facilities for a consultation. These individuals are tested for influenza disease, and positive cases are recognized as cases, while negative cases are identified as the controls. The effectiveness of the vaccine is measured by comparing the odds of infection between those vaccinated and unvaccinated [9,10]. The vaccine effect is measured by its effectiveness (VE), which is equal to
(2)$$\mathrm{VE}=\left(1-\mathrm{OR}\right)\times 100$$ where OR stands for the odds ratio. Several factors can affect the VE and introduce bias in the estimates of the VE. For example, VE can be seriously affected by the mismatch of the virus strains included in the vaccine and those in circulation in each vaccination season. The WHO has developed influenza surveillance and monitoring systems in order to understand which strains are circulating worldwide. Five reference centers located in the US, UK, Australia, Japan and China are responsible for collecting the information issued by each country and pinpointing the strains that are expected to be most prevalent in the following year, and these are the ones that should be in the next SIV [11]. The high rate of viral mutation, which includes fewer marked processes such as antigenic drift and profound changes called antigenic shift, mean that strains are not always as expected, to the notable detriment of the VE.

Previous vaccination may bias VE estimation. Natural infection and vaccination interfere with the individual’s immune system. On a theoretical level, it is expected that there may be some pre-existing immunity, either from previous infection or vaccination. Thus, resistance to the disease may be favored [12]. However, differences are not always significant [13]. The presence of comorbidities can affect VE. One of the groups at high risk of a severe influenza illness comprehends people with associated health problems. The influence of comorbidities can be analyzed in two ways: the ability of the interaction with the vaccine to be sufficiently robust for good protective capacity and the comorbidities as a determining influence on resistance to infection [14]. The type of substrate used for viral replication, using eggs or cultured cells may affect VE [15]. The time of the vaccination uptake is also relevant [16]. Finally, individual characteristics such as age [14,17], sex [14], conditions such as pregnancy [18] and even mood [19] at the time of the vaccine uptake seem to affect the VE.

Thus, RCTs and TNDs are used in different contexts. RCTs are the gold standard for licensing of use and TNDs are the main tool for monitoring the annual effectiveness of the SIVs [20,21]. However, these test designs apply different measures (VER and VE, respectively) and no relationship has been established between the results observed in RCTs and TNDs. This paper describes a systematic review and meta-analysis of RCTs and TNDs conducted to assess the VER and VE of SIVs in humans aged 15 to 64 years. Despite several papers describing performed reviews of the VER of the SIV or its effectiveness are available [22,23,24,25,26,27], information regarding VE is scarce. In this sense, the main objective of this work is to measure the effect of a vaccine assessed in RCTs and TNDs using a common measure: VE. Other information related to the individuals who participated in each study was also collected to identify possible factors that may influence VE.

## 2. Methods

This review adhered to the Preferred Reporting Items for Systematic Reviews and Meta-Analyses (PRISMA) criteria [28]. The proposed methodology for the systematic review was registered in the international prospective register of systematic reviews PROSPERO [CRD42023397974].

### 2.1. Search Strategy and Eligibility Criteria

An electronic search was conducted to identify all relevant studies. The literature search was performed in MEDLINE (via PubMed) and Cochrane’s library. The search in PubMed was performed using 
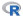
 4.3.0 software [29], package RISmed [30] available at https://www.r-project.org/. The last search date was March 2023. We also screened the reference list of included studies and relevant systematic reviews. The detailed search strategy is available in the register on PROSPERO. Only studies written in English were included to avoid any bias related to a mistranslation.

The included studies verified the following inclusion criteria:–Inclusion of sufficient information to compute the vaccine efficacy/effectiveness;–Articles published between 2013 and 2023;–Influenza was confirmed by a laboratorial method;–Articles that include the human population aged 15 to 65 years;–The participants in TNDs must have received a seasonal influenza vaccine at least 14 days before symptoms onset to be regarded as vaccinated;–Comparator was a placebo (for RCTs) or non-vaccinated (for TND);–Articles published in English.

TND studies were excluded if they reported only pooled data for more than one season.

### 2.2. Interventions

The intervention of interest was vaccination with one of the following seasonal influenza vaccines: trivalent inactivated (TIV), tetravalent inactivated (QIV) or live attenuated (LAIV). Monovalent and noncommercial vaccines were not considered.

### 2.3. Outcome Measure

The outcome of interest was vaccine efficacy or effectiveness on laboratory-confirmed influenza (any strain). PCR (polymerase chain reaction) and rapid virus detection tests were considered as possible methods for the confirmation of infection.

When studies reported both unadjusted and adjusted vaccine effectiveness, the adjusted figure was used in the results as it was considered the less biased estimate of the treatment effect.

Additional outcomes of interest were collected for subgroup analysis.

### 2.4. Data Collection and Analysis

Two authors of this review independently assessed the study eligibility by inspecting the title and abstract. All articles selected from the title/abstract reading were inspected for inclusion with a full-text review by both authors. The information of all selected papers was independently extracted to a form that included study design, participants, sample size, description of intervention, outcomes and quality assessment indicators. Discrepancies in study selection were resolved through consensus.

After the systematic review, a meta-analysis was performed using 
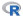
 software’s metafor package [31]. When any of the outcome measures were zero, the value of 0.5 was added. The associated confidence intervals are based on the logarithm transformation.

Forest plots were obtained to present a graphical overview (values of VE lower than −200% were omitted).

Between-study heterogeneity was assessed by Cochran’s Q and the $I^{2}$ statistic. A value higher than 0.75 was regarded as high heterogeneity and the pooled estimate was interpreted as low-certainty evidence. If the $I^{2}$ statistic was lower than 0.25, a fixed effects model was chosen. Otherwise, if the $I^{2}$ statistic was higher than 0.25 and lower than 0.75, a random effects model was used. Model estimation was performed using a restricted estimation maximum likelihood methodology with Knapp and Hartung adjustment. The adjusted estimates found in RCTs/TNDs were obtained using logistic regression. Thus, pooling was based on the log RR/OR and standard deviation, with the exponential of the pooled result re-expressed as VER/VE.

Funnel plots provided a visual assessment of possible publication bias. The trim and fill method and Egger’s test were applied to screen for possible publication bias [32].

### 2.5. Quality Assessment

Two authors independently assessed the included studies for risk of bias using validated critical appraisal tools. Inconsistencies were resolved by a third reviewer.

The Cochrane risk-of-bias tool for randomized trials (RoB 2) was used for RCTs [33]. Data were inputted in the RoB 2 Excel tool to implement them (available on riskofbiasinfo.org website, accessed on 1 March 2023).

TND studies were assessed for risk of bias using the ROBINS-I (Risk of Bias In Non-randomized Studies of Interventions) tool [34]. Results are presented in tabular form, with the agreed consensus of risk of bias for each of the seven included domains and the overall risk of bias for each study denoted by the highest risk of bias score in any singular domain, as per the ROBINS-I methodology. While unadjusted effectiveness was used where adjusted was not reported, there is clearly a risk of bias associated with the unadjusted estimate.

A sensitivity analysis was produced to control the effect of the high risk of bias studies.

## 3. Results

The search identified a total of 2993 records after removing duplicates (see Figure 1). The full text of 172 records was screened for eligibility, 99 of which were excluded. References of excluded studies are reported in Appendix A. A total of 123 studies from 73 papers were included in the meta-analysis. Of these studies, 9 are RCTs [35,36,37,38,39] and 114 are TNDs [40,41,42,43,44,45,46,47,48,49,50,51,52,53,54,55,56,57,58,59,60,61,62,63,64,65,66,67,68,69,70,71,72,73,74,75,76,77,78,79,80,81,82,83,84,85,86,87,88,89,90,91,92,93,94,95,96,97,98,99,100,101,102,103,104,105,106,107]. The selection process is detailed in the PRISMA flowchart (see Figure 1).

These articles comprised 86 studies from the Northern Hemisphere and 31 studies from the Southern Hemisphere. Six articles have information from countries in the Northern and in the Southern Hemisphere. In studies performed in more than one European country, the country is referred to as Europe (e.g., I-MOVE studies).

The main characteristics of the RCT studies reported in the articles are summarized in Table 1. Table 2 presents the characteristics of the TND studies.

The RCT sample size ranged from 85 to 7515 in the 9 studies. The reported VER ranged from −2% to 70%. The heterogeneity between studies is low: $I^{2}=9.9\%$ and Cochran’s Q=11.5 (*p*-value 0.18). The TIV was the most-used vaccine (seven studies). QIV and LAIV were used in only one study each. All individuals from one of the studies reported in [35] were HIV positive. Contact with the influenza virus in the study extracted from [39] was deliberately provoked. We find a match between the vaccine strains and the virus in circulation in six studies [36,37,38,39]. In two of the three studies where we verified a mismatch [35,37], it was possible to extract data for the matched strains. The studies extracted from [38] include only pregnant women.

The study by Liebowitz, D. et al. reported positive results for influenza for both ILI and non-ILI patients [39]. As our goal was to compare RCTs to TNDs, the efficacy concerning the influenza-positive illness was reported. The studies extracted from Mcbride, W.J. et al. [37] and Steinhoff, M.C. et al. [38] included two consecutive seasons.

The TND sample size ranged from 62 to 59,150. The TIV was the most-used vaccine (59 studies) followed by QIV (19 studies) in studies that used only 1 type of vaccine. Eighteen studies used both the TIV and QIV vaccines. LAIV was the only choice in three studies. Six studies used all three types of vaccines. The type of vaccine used in eight studies was unclear.

For the RCTs studies, the pooled adjusted VER is equal to 42.8% (95% CI: 30.5–52.9), which is similar to the pooled non-adjusted VER: 42.7% (95% CI: 30.1–53.0). In both models, $I^{2}=0$ (forest plots not shown).

Figure 2 presents a forest plot with the VE computed from the extracted crude values. This allows us to have a common measure to assess a vaccine performance regardless of the type of study (RCT or TND). The VE is represented by individual squares proportional to the precision of the estimates, and the horizontal lines represent the 95% CIs for each included study. The diamond indicates the pooled VE, which is equal to 45.3% (95% CI: 32.1–55.8).

For the TND studies, the reported VE ranged from −2% to 70%. The heterogeneity between studies is very high: $I^{2}=93.4\%$ and Cochran’s Q=247 (*p*-value < 0.001).

The pooled VE of the TIV is 48.3% (95% CI: 41.7–54.2) when there is a match between the strains included in the vaccine and the most prevalent in circulation, while it decreases to 40.1% (95% CI: 29.1–49.4) when there is a mismatch.

As for the QIV-type vaccine, studies show an overall effectiveness of 34.3% (95% CI: 29.6–38.7). When both the TIV and QIV vaccines are used in the same study, the effectiveness rate is 37.3% (95% CI: 24.5–47.8). In LAIV studies, the overall effectiveness is only 5.4% (95% CI: −20.7–25.9). Finally, when all three types of vaccines are used within the same study, the effectiveness is equal to 32.8% (95%CI: 12.3–48.5).

Forest plots of the adjusted VE of the TIV when vaccine strains match or mismatch are shown in Figure 3 and Figure 4. Figure 5, Figure 6, Figure 7 and Figure 8 show the results for the other type of vaccines (except for the only study that uses both QIV and LAIV). Further details about RCT and TND studies are given in Appendix A.

### 3.1. Risk of Bias Assessment

The RoB2.0 assessment indicates that one study [33] has a high risk of bias. A summary of the results is presented in Figure 9.

In the RCT studies, no evidence of publication bias is found through the funnel plot (Figure 10). Trim and fill methods and Egger’s test (*p*-value 0.19) do not identify any missing RCT study.

Figure 11 is the funnel plot for the TND studies. We found some evidence of asymmetry, which is confirmed by the trim and fill method (10 missing studies) and Egger’s test (*p*-value < 0.001). However, when restricting to studies where vaccine and circulating strains match, no missing studies were identified by the trim and fill method, and Egger’s test *p*-value increased to 0.04. When restricting to studies where vaccine and circulating strains mismatch, results were similar to the general case.

The ROBINS-I assessment tool for intervention was applied to papers involving TND studies. Most of the studies were at a serious/critical risk of bias as the vaccination status was not always based on the individual’s records (classification bias). Eleven papers were determined to have a low risk of bias. Fourteen studies failed to provide sufficient information to be classified in at least one of the seven domains analyzed by the tool. The overall results are summarized in Figure 12, and Appendix A presents the results by assessment domain.

### 3.2. Subgroup Analysis

A sensitivity analysis was performed by removing the high-bias RCT study [38]. The relative adjusted vaccine efficacy increased by 1.4% and the vaccine effectiveness by less than 0.5% As for the TND studies, those at a critical risk of bias were removed for the sensitivity analysis. The pooled VE showed non-significant variations. However, for the TIV vaccines when there was a mismatch between the circulating and vaccine strains, it decreased by nearly 10%.

Table 3 presents the vaccine effectiveness for the studies where the vaccine and circulating strains match and mismatch separately. A 15% increase in vaccine effectiveness was observed when the strains match. When our analysis is restricted to the TIV, the variation is even greater with an increase of more than 20%.

Table 4 presents a subgroup analysis for the TND studies. The TIV showed a better performance compared to studies that did not use a TIV (*p*-value 0.010). A match between the vaccine and circulating strains improves the VE by more than 10% (*p*-value 0.017). Confirming influenza by PCR results in a higher VE estimate (*p*-value 0.012).

No significant differences were found between pooled adjusted and non-adjusted VE estimates, studies that include individuals with severe symptoms and studies that included only ILI individuals, studies that included or not only inpatients, and studies performed in different hemispheres.

TIV pooled effectiveness observed in RCT studies is higher than the pooled value obtained from the TND studies when vaccine and circulating strains match, although the difference is not significant (*p*-value = 0.27, I^2^ = 17.4%). When there is a mismatch, the values are similar (around 40% in both cases).

## 4. Discussion

RCTs and TNDs are the most-used study designs to assess the performance of the SIV. Comparing RCT and TND estimates through a common measure (VE) is a relevant subject as the usefulness of TNDs is still discussed in the literature [10]. This explains the focus on individuals aged 15–64 years as they are not, in general, a high-risk group for severe influenza illness. Elderly people were excluded as it would increase the risk of dealing with results extracted from individuals with comorbidities. In addition, RCTs in the elderly population have another vaccine as a comparator [108,109]. Placebo is not used, as expected, because vaccination is recommended [110].

The number of RCT studies found was small. This fact limits the possibility of comparing RCT and TND studies except for TIV vaccines. The VE estimated by RCT studies is 10% higher than the VE estimated through the TNDs, although the difference is not significant. When there is a mismatch, similar values were obtained for both designs. It seems that TND studies are a reliable alternative for the assessment of a vaccine’s performance, as it is referred to in [111].

One of the main purposes of a meta-analysis is to compute pooled estimates. However, the pooled VE for the TND is not shown as it would be pointless. The VE of the individual studies varies over a wide range and the measure of the between-studies heterogeneity $I^{2}$ is close to 100%. However, it is possible to identify some reasons that explain this high heterogeneity.

From this review, the match between the vaccine and circulating strains arises as the most important factor. In TND studies, a difference of close to 10% was observed. This is in line with what was found in a systematic review of 2016 [112]. In elderly people, even greater differences were reported, between 20 and 30% [113,114]. When comparing TIV and QIV vaccines, we found higher effectiveness values for TIV vaccines. This is true for both RCT and TND studies. This was a surprising result as VE should increase with the number of strains included in the vaccine, although this was already observed in previous work on children [115]. Our understanding is that a match between the strains included in the vaccine and those that are predominantly circulating is the most influential factor. Hence, it is not relevant to have a high number of strains in a vaccine if they do not match the strains the vaccine aims to prevent. This also explains why the RCT study with a sample in which all individuals were HIV positive did not have a low VE, as there was a match. The relevance of the vaccine strains emerges as a key factor in effectiveness. This conclusion is supported by other meta-analyses whose results also point in this direction [116,117,118]. It is also interesting to observe that pooled VE obtained from adjusted and non-adjusted estimates are not significantly different. This leads one to believe that the impact of some of the confounding variables identified in the literature as influencing VE (e.g., prior vaccination) is limited, as some authors have already referred to [119].

PCR tests were used in most studies, although in some cases they were not the exclusive method for the detection of the influenza virus. It was not possible to compare the use of PCR tests with their non-use. Thus, it was only possible to compare the exclusive use of PCR tests with the combined use of more than one type of laboratory test. One of the alternative tests used was the rapid test, which has a lower sensitivity [120]. This lack of sensitivity might be the explanation for a significantly higher VE in the studies that used only PCR tests.

### Limitations of the Study

Despite the interesting results found herein, some limitations were evident. LAIV vaccines have very low effectiveness values. The number of studies (3) in which these vaccines are involved is low, so it is not possible to generalize the results. For instance, a VE equal to 44% was found in a systematic review reported in [115].

The TND is validated to assess vaccine effectiveness in outpatients but not in inpatients [57], which limited the possibility to compare disease severity with vaccine effectiveness. This may explain why only in a few studies individuals with acute respiratory infections were found. Thus, it was not possible to assess the impact of symptom severity.

As the number of studies reporting VE by age and strain was very low, it was not possible to assess the effect of the different strains in the VE estimation. Other limitations arise from the high number of TND studies at critical risk of bias and the lack of control over some variables, which could impact the reliability of the results. However, the majority of studies presented VE estimates adjusted for several confounding variables, although these variables were not always the same across the different studies.

## 5. Conclusions

This meta-analysis provides important insights into the effectiveness of influenza vaccines, highlighting the crucial role of the match between vaccine strains and those circulating in the population. The findings observed herein provide a basis for future research on the effectiveness of influenza vaccines and suggest that efforts should focus on improving the match between vaccine strains and those circulating in the population.

## Figures and Tables

**Figure 1 vaccines-11-01322-f001:**
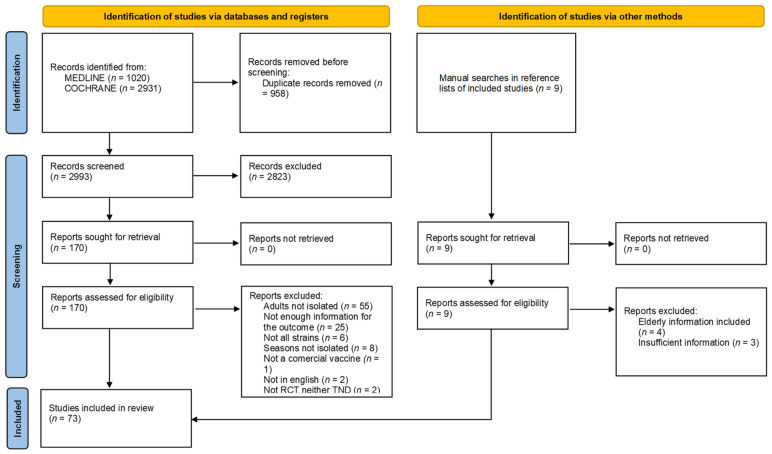
PRISMA study search flow diagram.

**Figure 2 vaccines-11-01322-f002:**
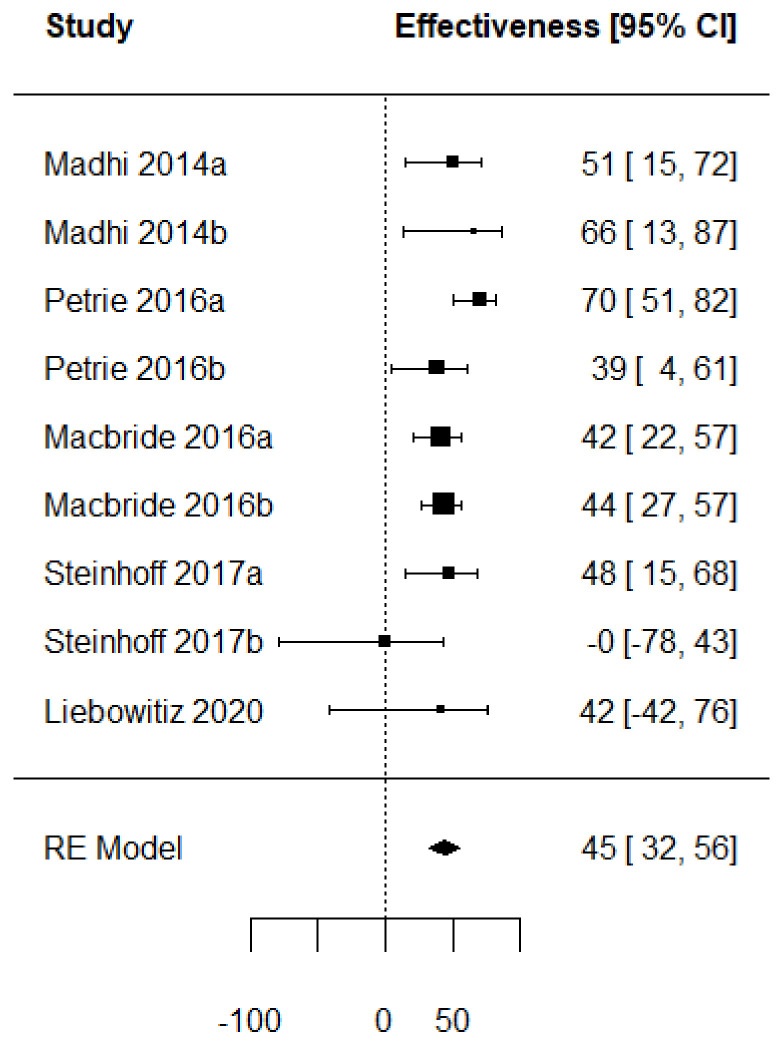
Forest plot of the vaccine effectiveness (RCT studies [35,36,37,38,39]).

**Figure 3 vaccines-11-01322-f003:**
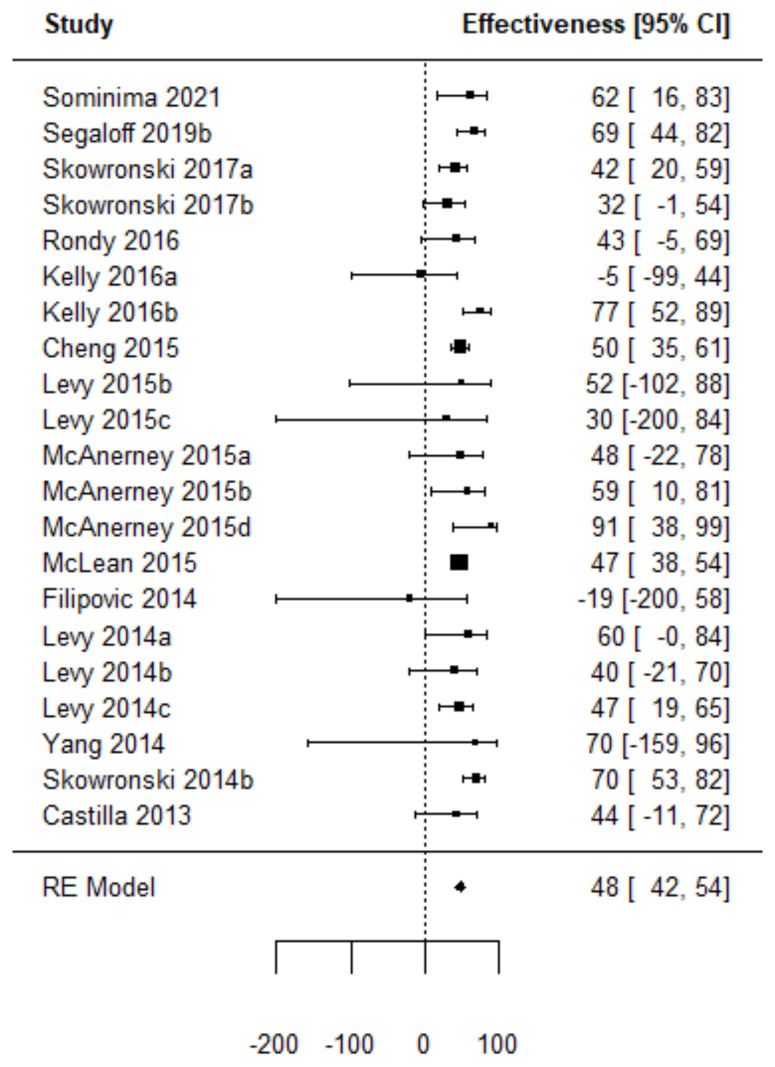
Forest plot of the vaccine effectiveness. (TND studies, TIV vaccine only, vaccine strains match circulating strains [45,57,78,87,90,92,93,94,97,98,100,101,104,107]).

**Figure 4 vaccines-11-01322-f004:**
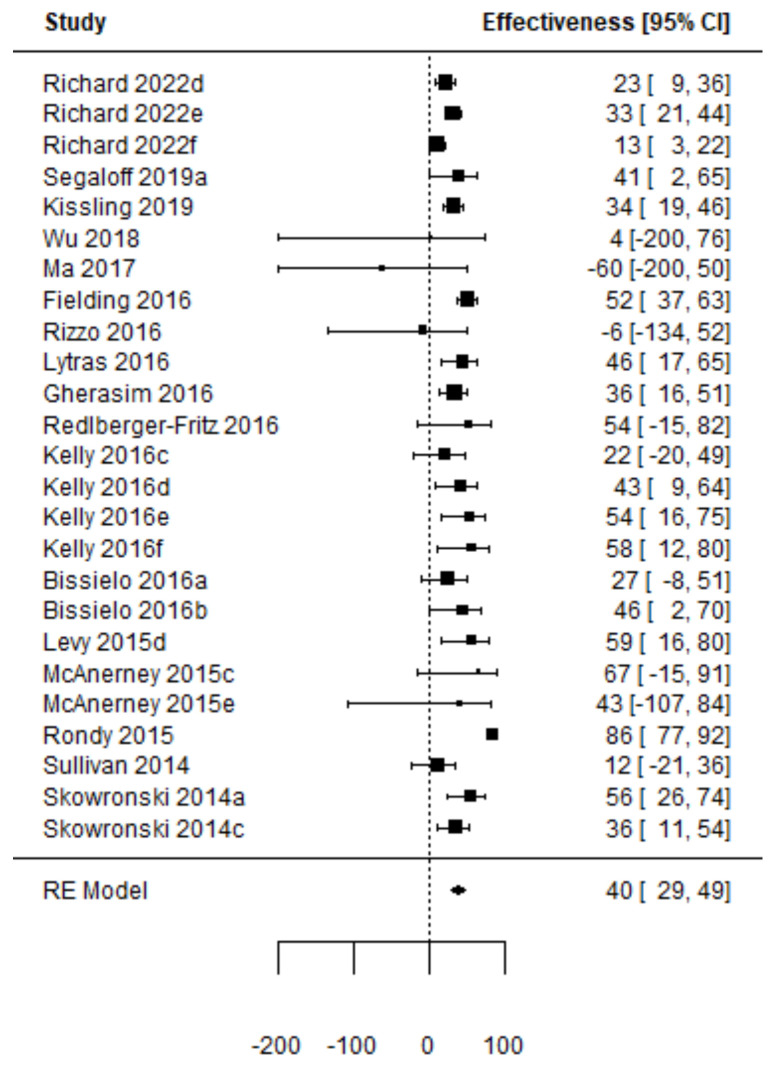
Forest plot of the vaccine effectiveness (TND studies, TIV vaccine only, mismatch between vaccine and circulating strains [44,57,59,73,80,83,85,86,88,89,90,91,93,94,95,96,102,103,105]).

**Figure 5 vaccines-11-01322-f005:**
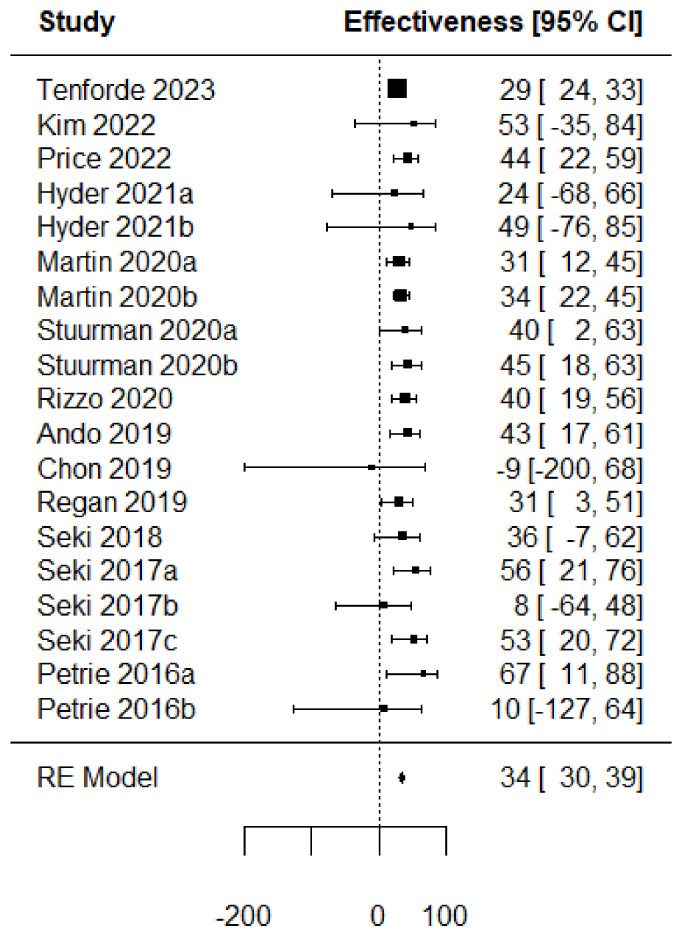
Forest plot of the vaccine effectiveness. (QIV vaccine only [41,42,43,46,50,51,52,56,64,72,81,84]).

**Figure 6 vaccines-11-01322-f006:**
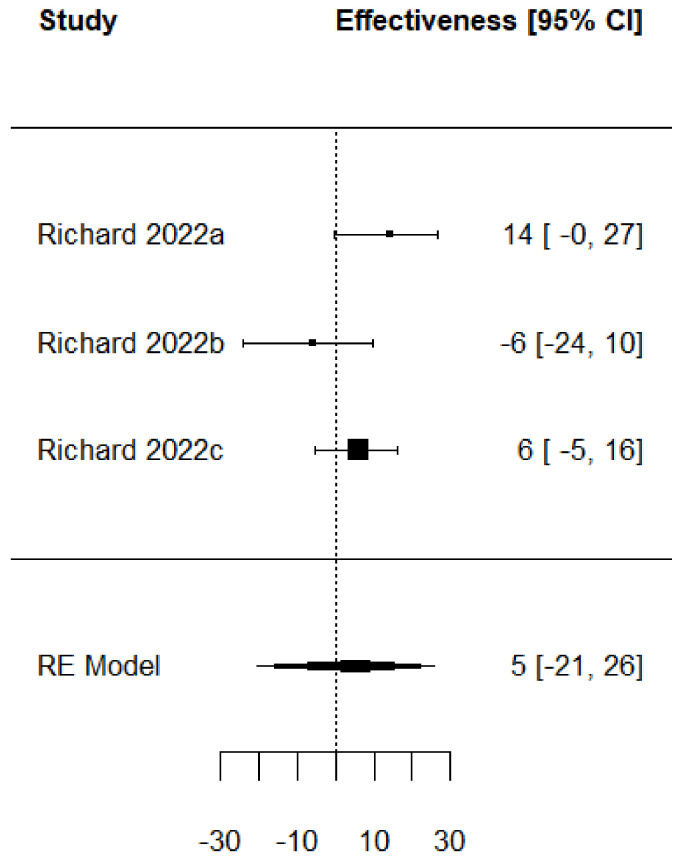
Forest plot of the vaccine effectiveness (LAIV vaccine only [44]).

**Figure 7 vaccines-11-01322-f007:**
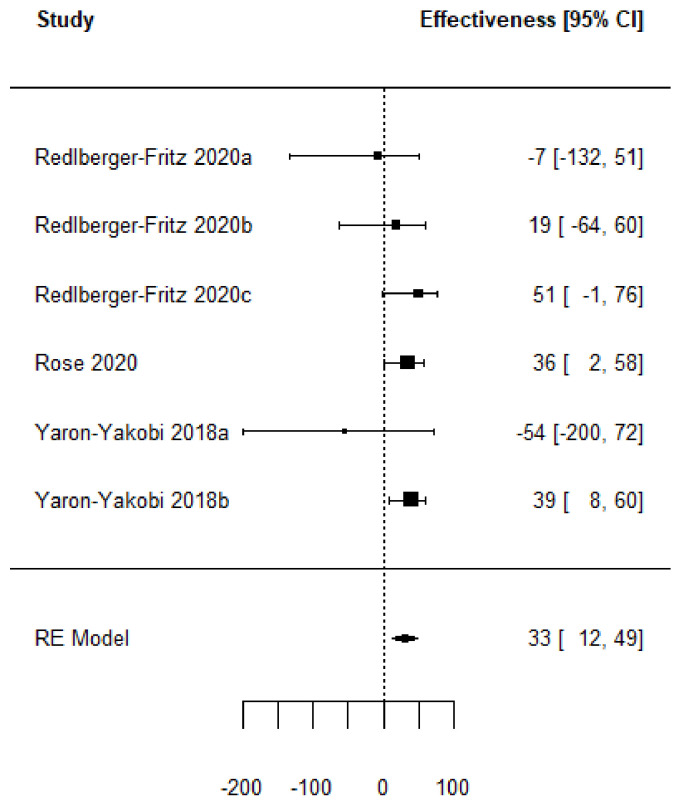
Forest plot of the vaccine effectiveness (TIV, QIV and LAIV simultaneously used [54,55,74]).

**Figure 8 vaccines-11-01322-f008:**
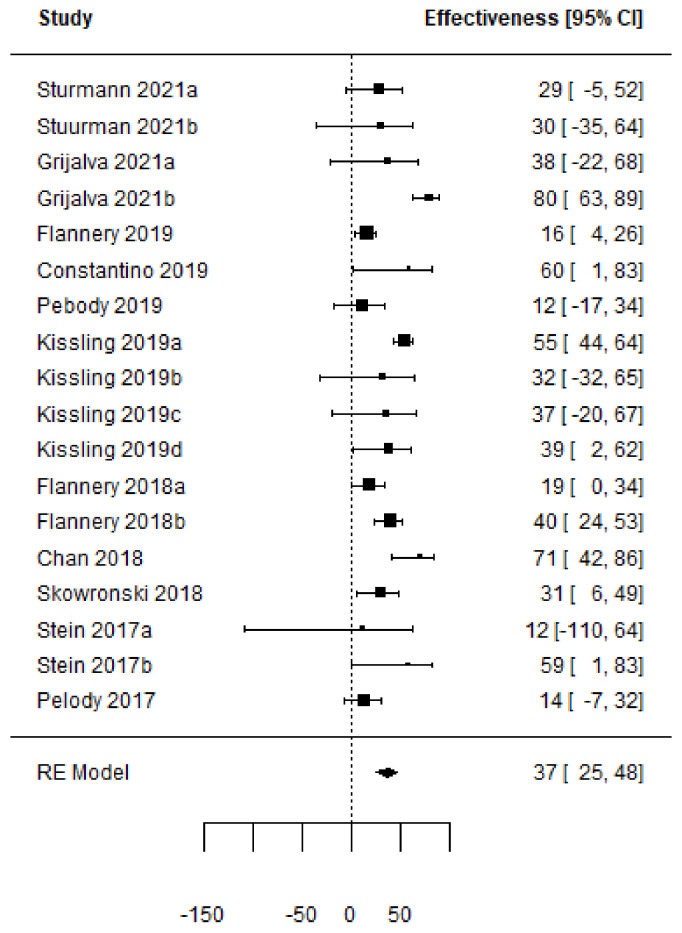
Forest plot of the vaccine effectiveness (TIV and QIV simultaneously used [47,48,58,61,62,63,70,71,75,76,77]).

**Figure 9 vaccines-11-01322-f009:**
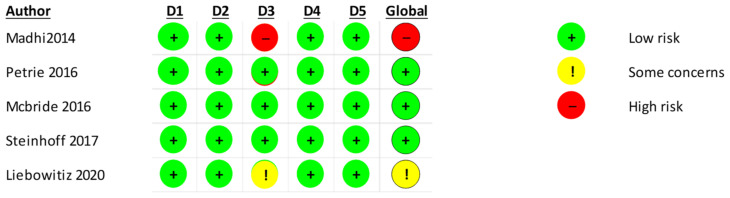
Rob 2.0 assessment of the included RCT studies [35,36,37,38,39] of the 5 domains. (D1: randomization process, D2: deviations from the intended interventions, D3: missing outcome data; D4: measurement of the outcome, D5: selection of the reported result).

**Figure 10 vaccines-11-01322-f010:**
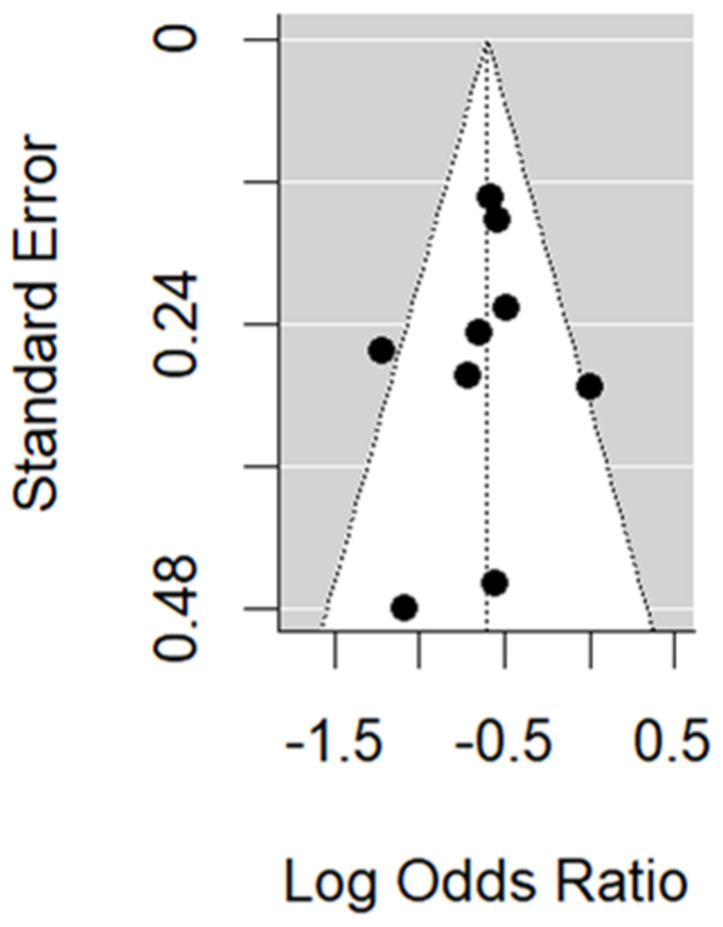
Funnel plot of the RCT studies.

**Figure 11 vaccines-11-01322-f011:**
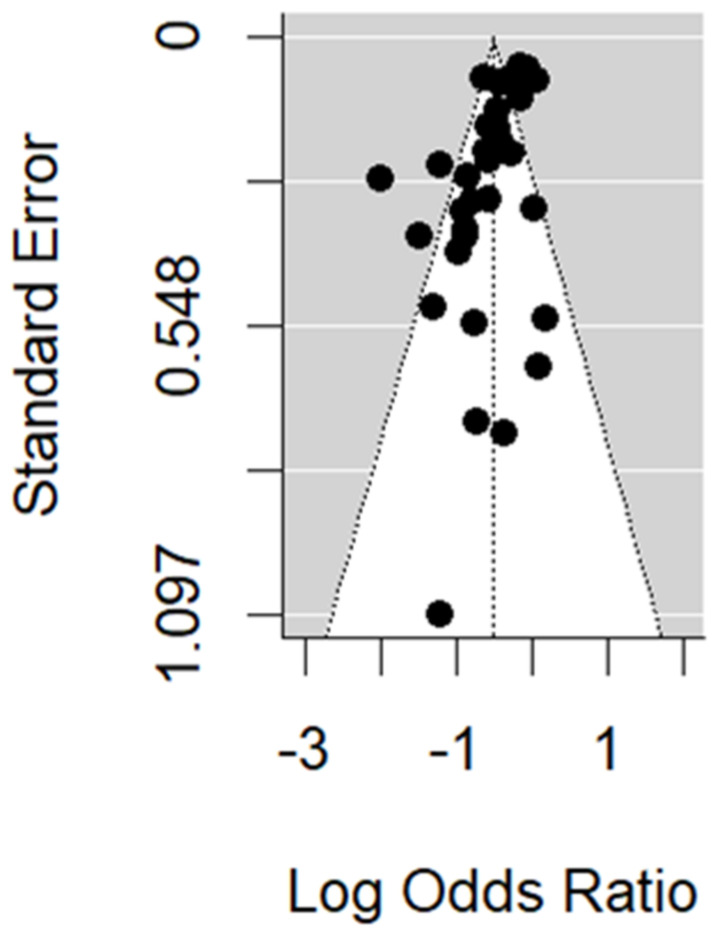
Funnel plot of the TND studies.

**Figure 12 vaccines-11-01322-f012:**
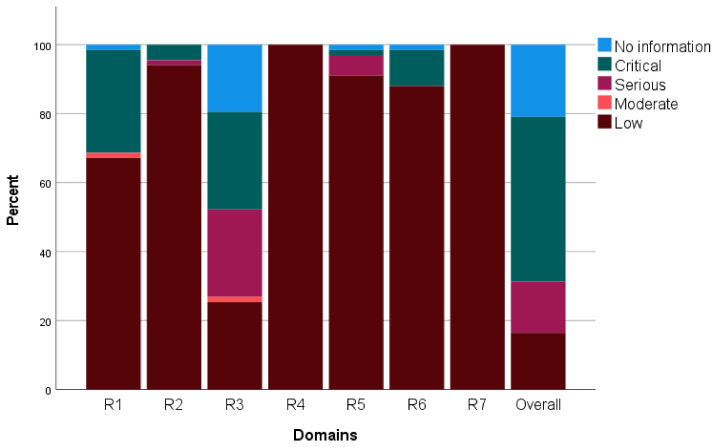
ROBINS-I overall assessment of the included TND studies and by domain. (R1: confounding, R2: selection, R3: classification, R4: deviation from intended interventions; R5: missing data; R6: outcome measurement; R7: reported results).

**Table 1 vaccines-11-01322-t001:** Summary of the included RCT studies.

Author	Country	Season	Vaccine	VE × 100 (95% CI)	*n*	Strain Match	Test
Madhi 2014 [35]	South Africa	2011–2012	TIV	51 (15–72)	2049	Mismatched	PCR
Madhi 2014 [35]	South Africa	2011–2012	TIV	66 (13–87)	188	Unclear	PCR
Petrie 2016 [36]	USA	2007–2008	TIV	70 (51–82)	1139	Matched	PCR
Petrie 2016 [36]	USA	2007–2008	LAIV	39 (4–61)	1138	Matched	PCR
Mcbride 2016 [37]	Australia	2008–2009	TIV	42 (22–57)	7515	Matched	PCR
Mcbride 2016 [37]	Australia	2008–2009	TIV	44 (27–57)	7334	Unclear	PCR
Steinhoff 2017 [38]	Nepal	2011–2012	TIV	48 (15–68)	3693	Unclear	PCR
Steinhoff 2017 [38]	Nepal	2012–2013	TIV	0 (−78–43)	3693	Matched	PCR
Liebowitiz 2020 [39]	USA	2012–2013	QIV	42 (−42–76)	2049	Matched	PCR

**Table 2 vaccines-11-01322-t002:** Summary of the included TND studies.

Author	Country	Season	Vaccine	VE × 100 (95% CI)	*n*	Strain Match	Test
Kissling 2023 [40]	Europe	2021–2022	QIV and LAIV	41 (25–64)	6876	Mismatched	PCR
Tenforde 2023 [41]	USA	2021–2022	QIV	29 (24–33)	59,150	Mismatched	PCR
Kim 2022 [42]	Canada	2021–2022	QIV	53 (−35–84)	176	Mismatched	PCR
Price 2022 [43]	USA	2021–2022	QIV	44 (22–59)	1850	Mismatched	PCR
Richard 2022 [44]	USA	2012–2013	LAIV	14 (0–27)	2580	Mismatched	PCR, RT, culture
Richard 2022 [44]	USA	2013–2014	LAIV	–6 (−24–10)	2613	Mismatched	PCR, RT
Richard 2022 [44]	USA	2014–2015	LAIV	6 (−5–16)	4715	Mismatched	PCR, RT
Richard 2022 [44]	USA	2012–2013	TIV	23 (9–36)	2311	Mismatched	PCR, RT
Richard 2022 [44]	USA	2013–2014	TIV	33 (21–44)	2517	Mismatched	PCR, RT
Richard 2022 [44]	USA	2014–2015	TIV	13 (3–22)	5043	Mismatched	PCR, RT
Sominina 2021 [45]	Russia	2018–2019	TIV	62 (16–83)	925	Matched	PCR
Hyder 2021 [46]	India	2017–2018	QIV	24 (−68–66)	547	Mismatched	PCR
Hyder 2021 [46]	India	2018–2019	QIV	49 (−76–85)	306	Mismatched	PCR
Stuurmann 2021 [47]	Europe	2019–2020	TIV and QIV	29 (−5–52)	1055	Mismatched	PCR
Stuurman 2021 [47]	Europe	2019–2020	TIV and QIV	30 (−35–64)	2041	Mismatched	PCR
Grijalva 2021 [48]	USA	2019–2020	TIV and QIV	38 (−22–68)	638	Mismatched	PCR
Grijalva 2021 [48]	USA	2019–2020	TIV and QIV	80 (63–89)	Unclear	Mismatched	PCR
Hu 2021 [49]	USA	2019–2020	Unclear	46 (36–55)	5817	Mismatched	PCR
Martin 2020 [50]	USA	2016–2017	QIV	31 (12–45)	2605	Matched	PCR
Martin 2020 [50]	USA	2017–2018	QIV	34 (22–45)	3524	Matched	PCR
Stuurman 2020 [51]	Europe	2018–2019	QIV	40 (2–63)	1095	Matched	PCR, RT
Stuurman 2020 [51]	Europe	2018–2019	QIV	45 (18–63)	2036	Matched	PCR, RT
Rizzo 2020 [52]	Italy	2018–2019	QIV	40 (19–56)	290	Mismatched	PCR
Qahtami 2020 [53]	Saudi Arabia	2018–2019	TIV	42 (14–64)	556	Unclear	PCR
Redlberger–Fritz 2020 [54]	Austria	2016–2017	QIV, TIV, aTIV, LAIV	−7 (−132–51)	492	Mismatched	PCR
Redlberger–Fritz 2020 [54]	Austria	2017–2018	QIV, TIV, aTIV, LAIV	19 (−64–60)	668	Mismatched	PCR
Redlberger–Fritz 2020 [54]	Austria	2018–2019	QIV, TIV, aTIV, LAIV	51 (−1–76)	614	Matched	PCR
Rose 2020 [55]	Europe	2019–2020	QIV, TIV, LAIV	36 (2–58)	13,630	Mismatched	PCR
Ando 2019 [56]	Japan	2018–2019	QIV	43 (17–61)	555	Unclear	RT
Segaloff 2019 [57]	USA	2014–2015	TIV	41 (2–65)	624	Mismatched	PCR
Segaloff 2019 [57]	USA	2015–2016	TIV	69 (44–82)	441	Matched	PCR
Flannery 2019 [58]	USA	2018–2019	TIV and QIV	16 (4–26)	5022	Mismatched	PCR
Kissling 2019 [59]	Europe	2016–2017	TIV	34 (19–46)	5840	Mismatched	PCR
Blanchette 2019 [60]	Canada	2010–2011	TIV	34 (20–40)	9288	Unclear	PCR
Constantino 2019 [61]	Italia	2018–2019	TIV and QIV	60 (1–83)	308	Probable	PCR
Pebody 2019 [62]	United Kingdom	2017–2018	TIV and QIV	12 (−17–34)	1896	Unclear	PCR
Kissling 2019 [63]	Denmark	2018–2019	TIV and QIV	55 (44–64)	5807	Mismatched	PCR
Kissling 2019 [63]	European Union	2018–2019	TIV and QIV	32 (−32–65)	1142	Mismatched	PCR
Kissling 2019 [63]	United Kingdom	2018–2019	TIV and QIV	37 (−20–67)	575	Mismatched	PCR
Kissling 2019 [63]	Denmark	2018–2019	TIV and QIV	39 (2–62)	727	Mismatched	PCR
Chon 2019 [64]	Japan	2015–2016	QIV	−9 (−200–68)	99	Unclear	PCR, RT
Molgaard–Nielsen 2019 [65]	Denmark	2010–2011	TIV	64 (29–82)	626	Unclear	Unclear
Regan 2019 [66]	Australia	2016	QIV	31 (3–51)	713	Mismatched	PCR
Showronski 2019 [67]	Canada	2017–2018	Unclear	63 (46–75)	946	Matched	PCR
Regan 2019 [68]	Australia	2012	TIV	46 (22–63)	825	Unclear	PCR
Regan 2019 [68]	Australia	2013	TIV	57 (26–75)	577	Unclear	PCR
Regan 2019 [68]	Australia	2014	TIV	60 (41–73)	1112	Unclear	PCR
Regan 2019 [68]	Australia	2015	TIV	50 (32–64)	1491	Unclear	PCR
Thompson 2018 [69]	USA	2010–2011	Unclear	72 (−5–93)	167	Unclear	PCR
Thompson 2018 [69]	USA	2011–2012	Unclear	47 (−98–86)	84	Unclear	PCR
Thompson 2018 [69]	USA	2012–2013	Unclear	23 (−85–68)	202	Unclear	PCR
Thompson 2018 [69]	USA	2013–2014	Unclear	51 (−30–82)	200	Unclear	PCR
Thompson 2018 [69]	USA	2014–2015	Unclear	24 (−189–47)	171	Unclear	PCR
Thompson 2018 [69]	USA	2015–2016	Unclear	40 (−33–72)	216	Unclear	PCR
Flannery 2018 [70]	USA	2017–2018	TIV and QIV	19 (0–34)	20,165	Matched	PCR
Flannery 2018 [70]	USA	2017–2018	TIV and QIV	40 (24–53)	1362	Matched	PCR
Chan 2018 [71]	China	2017–2018	TIV and QIV	71 (42–86)	383	Unclear	PCR
Seki 2018 [72]	Japan	2016–2017	QIV	36 (−7–62)	299	Matched	RT
Wu 2018 [73]	China	2016–2017	TIV	4 (−200–76)	6009	Mismatched	PCR
Yaron–Yakobi 2018 [74]	Israel	2014–2015	QIV, TIV, LAIV	−54 (−200–72)	417	Mismatched	PCR
Yaron–Yakobi 2018 [74]	Israel	2015–2016	QIV, TIV, LAIV	39 (8–60)	783	Mismatched	PCR
Skowronski 2018 [75]	Canada	2017–2018	TIV and QIV	31 (6–49)	895	Unclear	PCR
Stein 2017 [76]	Israel	2016–2017	TIV and QIV	12 (−110–64)	151	Matched	PCR
Stein 2017 [76]	Israel	2016–2017	TIV and QIV	59 (1–83)	165	Matched	PCR
Pelody 2017 [77]	United Kingdom	2017–2018	TIV and QIV	14 (−7–32)	1896	Mismatched	PCR
Skowronski 2017 [78]	Canada	2015–2016	TIV	42 (20–59)	1076	Matched	PCR
Skowronski 2017 [78]	Canada	2015–2016	TIV	32 (−1–54)	520	Matched	PCR
Kuliese 2017 [79]	Lithuania	2015–2016	TIV	61 (−345–97)	72	Unclear	PCR
Ma 2017 [80]	China	2014–2015	TIV	−60 (−200–50)	4990	Mismatched	PCR
Seki 2017 [81]	Japan	2013–2014	QIV	56 (21–76)	262	Unclear	RT
Seki 2017 [81]	Japan	2014–2015	QIV	8 (−64–48)	235	Unclear	RT
Seki 2017 [81]	Japan	2015–2016	QIV	53 (20–72)	427	Matched	RT
McAnerney 2016 [82]	South Africa	2015	TIV	54 (−14–82)	599	Unclear	PCR
Fielding 2016 [83]	Australia	2015	TIV	52 (37–63)	1492	Mismatched	PCR
Petrie 2016 [84]	USA	2014–2015	QIV	67 (11–88)	165	Mismatched	PCR
Petrie 2016 [84]	USA	2014–2015	QIV	10 (−127–64)	239	Mismatched	PCR
Rizzo 2016 [85]	Italy	2014–2015	TIV	−6 (−134–52)	1183	Mismatched	PCR
Lytras 2016 [86]	Greece	2014–2015	TIV	46 (17–65)	363	Mismatched	PCR
Rondy 2016 [87]	France, Italy e Spain	2013–2014	TIV	43 (−5–69)	305	Matched	PCR
Gherasim 2016 [88]	Spain	2014–2015	TIV	36 (16–51)	2957	Mismatched	PCR
Redlberger–Fritz 2016 [89]	Austria	2014–2015	TIV	54 (−15–82)	532	Mismatched	PCR
Kelly 2016 [90]	Australia	2011	TIV	−5 (−99–44)	227	Matched	PCR
Kelly 2016 [90]	Australia	2011	TIV	77 (52–89)	409	Matched	PCR
Kelly 2016 [90]	Australia	2012	TIV	22 (−20–49)	415	Mismatched	PCR
Kelly 2016 [90]	Australia	2012	TIV	43 (9–64)	460	Mismatched	PCR
Kelly 2016 [90]	Australia	2013	TIV	54 (16–75)	190	Mismatched	PCR
Kelly 2016 [90]	Australia	2013	TIV	58 (12–80)	258	Mismatched	PCR
Bissielo 2016 [91]	New Zealand	2015	TIV	27 (−8–51)	618	Mismatched	PCR
Bissielo 2016 [91]	New Zealand	2015	TIV	46 (2–70)	246	Mismatched	PCR
Cheng 2015 [92]	Australia	2014	TIV	50 (35–61)	1234	Matched	PCR
Levy 2015 [93]	Thailand	2009–2010	TIV	73 (26–90)	240	Mismatched	PCR
Levy 2015 [93]	Thailand	2010–2011	TIV	52 (−102–88)	62	Matched	PCR
Levy 2015 [93]	Thailand	2011–2012	TIV	30 (−200–84)	129	Matched	PCR
Levy 2015 [93]	Thailand	2012–2013	TIV	59 (16–80)	411	Mismatched	PCR
McAnerney 2015 [94]	South Africa	2010	TIV	48 (−22–78)	354	Matched	PCR
McAnerney 2015 [94]	South Africa	2011	TIV	59 (10–81)	548	Matched	PCR
McAnerney 2015 [94]	South Africa	2012	TIV	67 (−15–91)	749	Mismatched	PCR
McAnerney 2015 [94]	South Africa	2013	TIV	91 (38–99)	460	Matched	PCR
McAnerney 2015 [95]	South Africa	2014	TIV	43 (−107–84)	812	Mismatched	PCR
Rondy 2015 [96]	Europe	2012–2013	TIV	86 (77–92)	564	Mismatched	PCR
McLean 2015 [97]	USA	2012–2013	TIV	467	3307	Matched	PCR
Filipovic 2014 [98]	Croatia	2010–2011	TIV	−19 (−200–58)	240	Matched	PCR
Turner 2014 [99]	New Zealand	2014	TIV	59 (23–79)	190	Unclear	PCR
Turner 2014 [99]	New Zealand	2014	TIV	57 (28–74)	498	Unclear	PCR
Levy 2014 [100]	Australia	2010	TIV	60 (0–84)	355	Matched	PCR
Levy 2014 [100]	Australia	2011	TIV	40 (−21–70)	348	Matched	PCR
Levy 2014 [100]	Australia	2012	TIV	47 (19–65)	804	Matched	PCR
Yang 2014 [101]	China	2012–2013	TIV	70 (−159–96)	1246	Matched	Virus isolation
Sullivan 2014 [102]	Australia	2012	TIV	12 (−21–36)	926	Mismatched	PCR
Skowronski 2014 [103]	Canada	2011–2012	TIV	56 (26–74)	975	Mismatched	PCR
Skowronski 2014 [104]	Canada	2013–2014	TIV	70 (53–82)	562	Matched	PCR
Skowronski 2014 [105]	Canada	2012–2013	TIV	36 (11–54)	979	Mismatched	PCR
Kavanagh 2013 [106]	Scotland	2010–2011	TIV	100 (−349–100)	457	Matched	PCR
Castilla 2013 [107]	Spain	2011–2012	TIV	44 (−11–72)	650	Matched	PCR

**Table 3 vaccines-11-01322-t003:** Comparison of vaccine effectiveness according to the match between circulating and vaccine strains verified in the RCT studies (95% confidence intervals in brackets; a *p*-value lower than 0.05 is identified with *).

	Match	Mismatch	*p*-Value	$I^{2}$ (%)
All vaccines	55.4 (43.2,64.9)	39.3 (23.5, 51.9)	0.068	0.01
TIV	59.4 (46.4,69.2)	39.3 (23.5, 51.9)	0.035 *	0

**Table 4 vaccines-11-01322-t004:** Comparison of vaccine effectiveness according to several factors observed in TND studies. (95% confidence intervals are presented in brackets; a *p*-value lower than 0.05 or 0.01 is identified with * or **; ^a^ comparison between studies that only use the TIV vaccine and the other; ^b^ comparison between studies where vaccine and circulating strains match and mismatch, only for TIV studies; ^c^ comparison between studies that only use the QIV vaccine and the other; ^d^ comparison between studies that only use the LAIV vaccine and the other, ^e^ comparison between studies where vaccine and circulating strains match and mismatch, all studies; ^f^ comparison between studies that include individuals with severe symptoms and those that do not; ^g^ comparison between studies that include inpatient individuals and those that do not; ^h^ comparison between studies that confirm the presence of the influenza virus using only PCR and those that do not).

	Yes	No	*p*-Value	$I^{2}$ (%)
Adjusted estimate?	39.9 (30.5–47.9)	41.0 (36.8–44.9)	0.419	77.9
TIV? ^a^	44.9 (39.1–50.1)	30.3 (22.0–37.7)	0.010 *	76.9
Match circulating strains, TIV? ^b^	48.3 (41.7–54.2)	40.1 (29.1–49.4)	0.080	61.4
QIV? ^c^	34.3 (29.6–38.7)	42.7 (36.4–48.3)	0.454	80.8
LAIV? ^d^	5.4 (−20.7–25.9)	41.4 (37.1–45.4)	0.001 **	78.5
Match circulating strains? ^e^	45.1 (38.7–50.8)	35.1 (29.0–40.7)	0.017 *	69.8
Not only ILI? ^f^	38.7 (33.5–43.5)	43.9 (37.7–49.5)	0.176	73.9
Not only outpatients? ^g^	43.1 (33.2–51.6)	39.6 (35.1–43.8)	0.497	80.3
Only PCR? ^h^	42.7 (38.3–46.8)	29.7 (20.4–37.9)	0.012 *	76.3
Northern Hemisphere?	39.1 (34.4–43.4)	44.7 (36.1–52.2)	0.196	80.2

## Data Availability

The datasets used and/or analyzed during the current study are available from the corresponding author upon reasonable request.

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
