# Peer review of "Seasonal Influenza Vaccine Effectiveness in Persons Aged 15–64 Years: A Systematic Review and Meta-Analysis"

_vaccines, 2023, doi:10.3390/vaccines11081322_

Round 1
Reviewer 1 Report
Ensure compliance to the Journal format.

Author Response
We are pleased for your interest in our work. A few changes were performed and some information was added to give the readers a greater insight into how the systematic review and meta-analysis was performed.
Reviewer 2 Report
The introduction was very much like a lecture about how to assess vaccine effectiveness rather than how the systematic lit review was conducted. However, it is a good review for people new to the area. However, it reduced the detail available about how the SR and MA were conducted.
Further, the neglect of information about cohort studies was very apparent.
LINE COMMENT
16 if not all types of observational design studies, indicate only test-negative (and spell out the first time you use the initialization)
24 This conclusion is surprising given the results in the abstract. I suggest including more information in the results section to support this.
53-54 Cohort studies are often used. Suggest saying “For monitoring the annual effectiveness when laboratory confirmation is required, the TND is used.”
58 Remove the sentence “The use the VER to assess the effect of the vaccine is no longer advised.”
47 & 59 Use the word “that” rather than the word “which” (more correct English usage)
61-62 Remove the sentence “A value of 100% in (2) indicates prevention of all cases whereas 0% indicates prevention of no cases of influenza.”
105-109 Limiting the search to PubMed is limiting the number of manuscripts attained. They also neglected to search the clinical trials registries. These are vital for RCTs.
150-151 Incomplete sentence
185-186 I suggest moving information from 222-228 up here. You are giving me the results of the meta-analysis before telling me about the studies included thereof. Please clarify which studies are included where and then give the results. The figures could be more effective if done the same way.
224-225 I suggest contacting the authors to clarify the vaccines used (although harder with TND).
229-231 Provide 95% CI for both estimates
RoB If the risk of bias is high/critical, why would you include the study results? Why bother doing the RoB assessment if you aren’t going to use it? I see that you included information without the one highly biased RCT, but didn’t do the same for the TND.
Tables/Figures
Please align to left or right. Centering is hard to read
Not sure why there are lines in table 2 under certain rows
Figures are not well labelled and need footers to define abbreviations/initializations
I assume that Figure 3 is for the RCTs. Figure 3 says it is for RCTs, but the labels indicate TND studies, I think. Figure 1 follows Figure 3, apparently….I assume it is TNDs, but not sure what grouping...very confusing.
I think table 3 could be explained in text without the table.
I cannot, even with reading the text, understand what you are trying to show in Table 4.
Also
Supplementary tables in systematic reviews typically include other information. I suggest the authors review the PRISMA criteria and report on these criteria.
It would likely have good impact on the paper to have an infection control physician assist in the writing of the paper. Perhaps also someone who has conducted a systematic review.
Quality of English is acceptable.
The format and layout of the manuscript is poor
Author Response
Dear reviewer, we hope our changes in the manuscript meet your expectations. Please find below a reply to each of your comments.
Comment: The introduction was very much like a lecture about how to assess vaccine effectiveness rather than how the systematic lit review was conducted. However, it is a good review for people new to the area. However, it reduced the detail available about how the SR and MA were conducted.
Reply: We described the SR and MA in the Methods section. A new supplementary table was added to identify the excluded studies and the reason for exclusion.
Comment: Further, the neglect of information about cohort studies was very apparent.
Reply: Indeed the cohort studies were neglected. The objective of the investigation that triggered the beginning of this research work was the comparison of the TND with the RCT, given the wide use of the TND and the existing discussion in the literature about how different factors can bias the results. We recognize the relevance of cohort studies and that they may in the future be part of a more comprehensive comparative analysis of observational studies with RCTs.
LINE COMMENT
16 if not all types of observational design studies, indicate only test-negative (and spell out the first time you use the initialization)
Reply: Done.
24 This conclusion is surprising given the results in the abstract. I suggest including more information in the results section to support this.
Reply: The information about the non-adjusted VE was deleted as it was not relevant for the main conclusions. Information supporting the main conclusion was added.
53-54 Cohort studies are often used. Suggest saying “For monitoring the annual effectiveness when laboratory confirmation is required, the TND is used.”
Reply: Done
58 Remove the sentence “The use the VER to assess the effect of the vaccine is no longer advised.”
Reply: Done
47 & 59 Use the word “that” rather than the word “which” (more correct English usage)
Reply: Done
61-62 Remove the sentence “A value of 100% in (2) indicates prevention of all cases whereas 0% indicates prevention of no cases of influenza.”
Reply: Done
105-109 Limiting the search to PubMed is limiting the number of manuscripts attained. They also neglected to search the clinical trials registries. These are vital for RCTs.
Reply: The search was not limited to PubMed as it would in fact limit the number of manuscripts attained. We’ve searched in Cochrane’s library that allows access to other databases such as ClinicalTrials.gov that provides access to clinical studies from around the world.
150-151 Incomplete sentence
Reply: The sentence was rewritten.
185-186 I suggest moving information from 222-228 up here. You are giving me the results of the meta-analysis before telling me about the studies included thereof. Please clarify which studies are included where and then give the results. The figures could be more effective if done the same way.
Reply: Done
224-225 I suggest contacting the authors to clarify the vaccines used (although harder with TND).
Reply: The number of RCTs found was reduced but this information was available in all. In TND, the number of studies in which vaccines are not identified is small and may be motivated using different vaccines. Therefore, we decided to continue the study, albeit with this limitation.
229-231 Provide 95% CI for both estimates
Reply: Done
RoB If the risk of bias is high/critical, why would you include the study results? Why bother doing the RoB assessment if you aren’t going to use it? I see that you included information without the one highly biased RCT,but didn’t do the same for the TND.
Reply: Indeed the impact of the high risk of TND studies should have been reported. Results on this topic were added to the text.
Tables/Figures
Please align to left or right. Centering is hard to read
Not sure why there are lines in table 2 under certain rows
Figures are not well labelled and need footers to define abbreviations/initializations
I assume that Figure 3 is for the RCTs. Figure 3 says it is for RCTs, but the labels indicate TND studies, I think. Figure 1 follows Figure 3, apparently….I assume it is TNDs, but not sure what grouping...very confusing.
Reply: Figures and Tables were edited, and several labels were rewritten.
I think table 3 could be explained in text without the table.
Reply: Table 3 actually includes little information that could be presented in the body of the text. However, we chose to place it in a table format in order to highlight the values ​​presented, as they support one of the main conclusions of the study.
I cannot, even with reading the text, understand what you are trying to show in Table 4.
Reply: Additional information was added to the Table 4 label and to the text.
Also
Supplementary tables in systematic reviews typically include other information. I suggest the authors review the PRISMA criteria and report on these criteria.
Reply: An additional supplementary table was added with the excluded studies and the reason for exclusion
The format and layout of the manuscript is poor
Reply: Issues detected in the format and layout were corrected
Reviewer 3 Report
The authors conducted systematic research on the efficacy of vaccines. The authors operated by separating and comparing RCT and TND estimates of the VE. The criteria for the systematic review and those for the meta-analysis are correctly followed. Some manuscript improvements are suggested.
1. The abstract is not composed according to editorial standards. Authors should remove the section headings "Introduction, Methods..."
2. Keywords repeat the words in the title. This makes them practically useless. I advise authors to choose other terms, which express the meaning of the work, but are not given in the title.
3. Among the limitations of the study should be considered the lack of references to the extensive literature on the subject. The topic is very popular. Searching PubMed under the heading "influenza vaccination effectiveness" limiting to meta-analysis and systematic review, there are 648 results. There are 361 meta-analyses. The authors should cite some of these studies and explain why their study was necessary. The results obtained in this study should be compared with the results obtained from previous studies.
Author Response
Dear reviewer, we hope our changes in the manuscript meet your expectations. Please find below a reply to each of your comments.
Comment: The authors conducted systematic research on the efficacy of vaccines. The authors operated by separating and comparing RCT and TND estimates of the VE. The criteria for the systematic review and those for the meta-analysis are correctly followed. Some manuscript improvements are suggested.
- The abstract is not composed according to editorial standards. Authors should remove the section headings "Introduction, Methods..."
Reply: Done
- Keywords repeat the words in the title. This makes them practically useless. I advise authors to choose other terms, which express the meaning of the work, but are not given in the title.
Reply: The chosen keywords were adjusted to avoid repetitions.
- Among the limitations of the study should be considered the lack of references to the extensive literature on the subject. The topic is very popular. Searching PubMed under the heading "influenza vaccination effectiveness" limiting to meta-analysis and systematic review, there are 648 results. There are 361 meta-analyses. The authors should cite some of these studies and explain why their study was necessary.The results obtained in this study should be compared with the results obtained from previous studies.
Reply: Results of other meta-analysis are now referred and have improved the Discussion of the paper.
Reviewer 4 Report
In this manuscript by Martins et al., the author present a systematic review and meta-analysis on the effectiveness of the seasonal influenza vaccines, which include the trivalent inactivated, tetravalent inactivated, and the live attenuated forms. The authors focus their review and meta-analysis on randomized controlled trials and observational studies. Not surprisingly, the authors found that the most critical factor influencing the efficacy of influenza vaccines is the degree of match between the vaccine antigens and the infectious strains in circulation. Specific comments are provided below.
(1) The title of the manuscript is misleading ("Seasonal Influenza Vaccine Effectiveness in Adults...") since the authors' study population included persons aged 15-64 years.
(2) It is recommended that the authors use a professional English proofreader to clean up expression/grammatical errors. For example, in the Methods subsection in the Abstract, line 17-18 (starting with "An electronic search...") is not a complete sentence. In another example, the semicolon in line 126 should be a colon (:).
(3) The Introduction is rather long and could be better organized into well-developed paragraphs of 5 sentences or more instead of several paragraphs consisting of 3 sentences.
(4) The formatting in Table 2 is poor. Please annotate what "e" means in the table and remove extraneous underlines. Under the "Test" category, what is the meaning of "tesrapi" after RT. All acronyms used in the table should be defined.
(5) The Forest plot figure designated as "Figure 1" is mislabeled and should be "Figure 4".
(6) The resolution quality of the Forest plot figures is not consistent and should be improved.
See comments on the quality of English above.
Author Response
Dear reviewer, we hope our changes in the manuscript meet your expectations. Please find below a reply to each of your comments.
Comment: In this manuscript by Martins et al., the author present a systematic review and meta-analysis on the effectiveness of the seasonal influenza vaccines, which include the trivalent inactivated, tetravalent inactivated, and the live attenuated forms. The authors focus their review and meta-analysis on randomized controlled trials and observational studies. Not surprisingly, the authors found that the most critical factor influencing the efficacy of influenza vaccines is the degree of match between the vaccine antigens and the infectious strains in circulation. Specific comments are provided below.
(1) The title of the manuscript is misleading ("Seasonal Influenza Vaccine Effectiveness in Adults...") since the authors' study population included persons aged 15-64 years.
Reply: Done. Title changed.
(2) It is recommended that the authors use a professional English proofreader to clean up expression/grammatical errors. For example, in the Methods subsection in the Abstract, line 17-18 (starting with "An electronic search...") is not a complete sentence. In another example, the semicolon in line 126 should be a colon (:).
Reply: Done.
(3) The Introduction is rather long and could be better organized into well-developed paragraphs of 5 sentences or more instead of several paragraphs consisting of 3 sentences.
Reply: Done. Introduction re-organized.
(4) The formatting in Table 2 is poor. Please annotate what "e" means in the table and remove extraneous underlines. Under the "Test" category, what is the meaning of "tesrapi" after RT. All acronyms used in the table should be defined.
Reply: The letter “e” was replaced by “and”. The acronyms are now defined.
(5) The Forest plot figure designated as "Figure 1" is mislabeled and should be "Figure 4".
Reply: Done.
(6) The resolution quality of the Forest plot figures is not consistent and should be improved.
Reply: The quality of the Forest plots was improved.
Round 2
Reviewer 2 Report
Table 2 has duplicate observations. Tables 1 & 2 should be expanded to include more detail so the reader can easily review the variables that differ between them (including the VE/VER) and number of participants/patients.
Figure 2. Change wording in title & label to "efficacy" as these are RCTs; Also include footer/explain what RE model stands for
Figure 3 & 8. Skowronski is misspelled
Line 237 & 238 & 239: are you discussing VE or VER? Please be consistent
Line 244-245: VE is lower but with much overlap in confidence bands/intervals
Line 297-299: You are talking about RCTs but include vaccine effectiveness. Please clarify.
Tables 3 & 4: you are, in effect, doing "multiple testing" by including studies in more than one comparison. Delete any indication that a p-value of <0.10 is significant.
Discussion: contains results. Please move them to the results section (lines 340-342
The discussion is rambling; it is not focused and polished. A rewrite would make it more applicable and easier to follow for the reader.
The limitations should include issues regarding confounding and bias possible with TNDs.
English is improved over the first draft but still requires some work to make it easier to read and understand.
Author Response
Dear reviewer, please find below a reply to each one of your comments. They were useful to improve our work.
Comment: Table 2 has duplicate observations. Tables 1 & 2 should be expanded to include more detail so the reader can easily review the variables that differ between them (including the VE/VER) and number of participants/patients.
Reply: Duplicate observations removed. Sample size and VE added in both Tables. In Table 2, the adjusted VE was added when available.
Comment: Figure 2. Change wording in title & label to "efficacy" as these are RCTs; Also include footer/explain what RE model stands for
Reply: The meaning of RE was added. Vaccine efficacy is provided in Supplementary Table 2. We decide to present the computed vaccine effectiveness using each study raw data because one of the purposes of the work is to compare TND and RCT. Vaccine effectiveness is used in order to have a common measure. This option is now clearer in the text.
Comment: Figure 3 & 8. Skowronski is misspelled
Reply: Done
Comment: Line 237 & 238 & 239: are you discussing VE or VER? Please be consistent
Reply: A sentence in line 239 was moved forward to make it clear when VE or VER is being discussed.
Comment: Line 244-245: VE is lower but with much overlap in confidence bands/intervals
Reply: That’s true and the difference is non-significant as stated later when analysing Table 4. We left the comparison between groups of studies only to the section Subgroup analysis.
Comment: Line 297-299: You are talking about RCTs but include vaccine effectiveness. Please clarify.
Reply: As previously said in a previous reply, a sentence was added to make it clearer why to use the vaccine effectiveness when discussing RCTs. The purpose of the paper is not to estimate the pooled vaccine efficacy but to analyze if the results are similar when the type of study.
Comment: Tables 3 & 4: you are, in effect, doing "multiple testing" by including studies in more than one comparison. Delete any indication that a p-value of <0.10 is significant.
Reply: In each test non-overlapping groups of studies are used. Thus, there are no issues regarding the tests involved. Although the criteria for defining the subgroups is clearly defined for each test, we acknowledge that having to compare, for instance, studies with only ILI patients to the others is not the best way to assess the severity of the symptoms effects on effectiveness. However, that’s what the available data allowed us to do.
The indication of a p-value <0.10 was deleted.
Comment: Discussion: contains results. Please move them to the results section (lines 340-342
Reply: Done
Comment: The discussion is rambling; it is not focused and polished. A rewrite would make it more applicable and easier to follow for the reader.
Reply: Some changes were performed in Discussion to make it clear.
Comment: The limitations should include issues regarding confounding and bias possible with TNDs.
Reply: Done
Comment: English is improved over the first draft but still requires some work to make it easier to read and understand.
Reply: Some improvements were performed throughout the text
Reviewer 3 Report
The authors conveniently revised the manuscript
Author Response
We are pleased to have responded to the questions raised by you.